

# HAHap: a read-based haplotyping method using hierarchical assembly

Yu-Yu Lin[1], Ping Chun Wu[2], Pei-Lung Chen[3], Yen-Jen Oyang[1] and Chien-Yu Chen[4]

[1] Department of Graduate Institute of Biomedical Electronics and Bioinformatics, National Taiwan University, Taipei, Taiwan
[2] Taipei Blood Center, Taiwan Blood Services Foundation, Taipei, Taiwan
[3] Graduate Institute of Medical Genomics and Proteomics, College of Medicine, National Taiwan University, Taipei, Taiwan
[4] Department of Bio-Industrial Mechatronics Engineering, National Taiwan University, Taipei, Taiwan

## ABSTRACT

**Background:** The need for read-based phasing arises with advances in sequencing technologies. The minimum error correction (MEC) approach is the primary trend to resolve haplotypes by reducing conflicts in a single nucleotide polymorphism-fragment matrix. However, it is frequently observed that the solution with the optimal MEC might not be the real haplotypes, due to the fact that MEC methods consider all positions together and sometimes the conflicts in noisy regions might mislead the selection of corrections. To tackle this problem, we present a hierarchical assembly-based method designed to progressively resolve local conflicts.

**Results:** This study presents HAHap, a new phasing algorithm based on hierarchical assembly. HAHap leverages high-confident variant pairs to build haplotypes progressively. The phasing results by HAHap on both real and simulated data, compared to other MEC-based methods, revealed better phasing error rates for constructing haplotypes using short reads from whole-genome sequencing. We compared the number of error corrections (ECs) on real data with other methods, and it reveals the ability of HAHap to predict haplotypes with a lower number of ECs. We also used simulated data to investigate the behavior of HAHap under different sequencing conditions, highlighting the applicability of HAHap in certain situations.

Corresponding author
Chien-Yu Chen,
chienyuchen@ntu.edu.tw

## INTRODUCTION

Haplotype phasing, also known as haplotyping, is the process of resolving precise haplotypes. A haplotype describes serial genetic variants that co-occur on a single chromosome. Characterization of haplotypes is essential in various research problems, including allelic expression (*Castel et al., 2016*), linkage analysis, association studies (*Nalls et al., 2014*; *Ripke et al., 2013*), population genetics (*Sankararaman et al., 2014*; *Schiffels & Durbin, 2014*) and clinical genetics (*Zanger & Schwab, 2013*). Many studies

have proposed experimental or computational phasing approaches to resolve haplotypes (*Browning & Browning, 2011*; *Snyder et al., 2015*). Advances in sequencing technologies have facilitated faster and cheaper resolution of haplotypes, while increasing the need for efficient and effective computational methods.

Several computational phasing approaches have been proposed in recent years. Genetic phasing makes use of related individuals to achieve better precision but can only be adopted when pedigrees are available (*O'Connell et al., 2014*). Population phasing uses genotyping data of a large cohort to infer haplotypes (*Glusman, Cox & Roach, 2014*). However, this approach is only applicable to well-known variants in a population. Read-based phasing utilizes reads spanning at least two heterozygous variants to infer haplotypes (*Edge, Bafna & Bansal, 2017*; *Garg, Martin & Marschall, 2016*; *Mazrouee & Wang, 2014*; *Pirola et al., 2016*). With the appearance of long-read sequencing technologies, where longer reads could possibly cover more heterozygous variants, read-based phasing has increasingly become more appealing. However, since long-read sequencing still suffers high cost at this moment, it may not deliver as high coverage as short-read sequencing. In this regard, individual phasing based on short reads with a sequencing depth of at least $30\times$ remains competitive nowadays. Even though individual phasing based on short reads cannot deliver as high quality as the long-read or population-based solutions (*Choi et al., 2018*), whole-genome sequencing (WGS) becomes more and more critical and frequently used in medical research and precision medicine (*Ellingford et al., 2016*; *Glusman, Cox & Roach, 2014*; *Luukkonen et al., 2018*; *Sousa-Pinto et al., 2016*; *Stavropoulos et al., 2016*).

An important computational model developed for read-based phasing is the minimum error correction (MEC) model (*Lancia et al., 2001*). The MEC approach corrects putative errors in the reads by changing the alleles that cause conflicts. Conflicts are the alleles in reads that do not support the predicted haplotypes. In the MEC approach, conflicts are considered as errors, which might be sequencing errors or alignment errors. The MEC problem was proven to be NP-hard (*Lippert et al., 2002*). In this regard, methods delivering the optimal solutions were usually time-consuming (*Chen, Deng & Wang, 2013*; *He et al., 2010*), and are considered impractical in whole-genome phasing with a sequencing depth of $30\times$ or more. Here, we used an Illumina WGS sample downloaded from the Genome in a Bottle Consortium (GIAB) as an example. This sample (sample ID: NA24143) consisted of $2 \times 250$ paired-end reads, with a sequencing depth of about $40\times$. By using a 48-core machine with Intel Xeon E5-2683 CPUs and 384 GB of memory, an ILP method, HapAssembly, took more than 150 h to complete the phasing, in both general and all-heterozygous modes. This is impractical for an application that contains more than hundreds of WGS samples.

In this regard, many computational methods have been proposed to speed up the phasing performance. WhatsHap (*Garg, Martin & Marschall, 2016*) is a well-known phasing tool that down-samples the read sets first and uses a dynamic programming fixed parameter tractability algorithm to solve a weighted MEC problem faster. While WhatsHap delivers optimal solutions, several heuristic methods have been proposed to find haplotypes more efficiently (*Aguiar & Istrail, 2012*; *Edge, Bafna & Bansal, 2017*;
*Xie, Wang & Chen, 2015*). HapCut2 is a maximum likelihood estimation heuristic method that uses max-cut computation to search a subset of variants such that changing haplotypes on those variants is possible to achieve a greater likelihood. The procedure is repeated until no further improvements are possible, eventually leading to a near-optimal approach of MEC.

Although MEC is regarded as the state-of-the-art strategy, there is room for improvement. First, the quality of the reconstructed haplotypes may be severely affected by sequencing and alignment errors. Conceptually, MEC methods search solutions in a global manner, which works well if there are relatively fewer noises than signals. Second, read-based phasing struggles when handling regions with dense variants, which is computationally infeasible for many existing methods. The execution time of a read-based phasing method is highly related to the sequencing coverage and the number of variants. In the future, longer reads from third-generation sequencing technologies will necessitate a more time-efficient method, since longer reads involve more variants in the problem.

This study aims to develop a heuristic read-based haplotyping method based on a hierarchical assembly algorithm. Our method is not designed to solve the MEC problem. Instead, it attempts to eliminate the influence of noises through iteratively considering the most reliable information from variant pairs. We developed an adjusted multinomial probabilistic metric for evaluating the reliability of a variant pair, and the derived scores guide the assembly process. Once only the pairs with low scores remain, we accept a local MEC search method to resolve the haplotypes of this local region, with a much smaller search space than what used to be. The evaluation was performed using the haplotype prediction of the Ashkenazim trio by $10\times$ Genomics, which uses a barcoding technique followed by pooled short read sequencing to resolve haplotypes (*Porubsky et al., 2017*; *Zheng et al., 2016*). This study compares the proposed method with an exact method tool, WhatsHap, and a heuristic method, HapCut2, on both real and simulated data. We also generated simulated data in different situations based on the hg19 genome in order to study the applicability of HAHap. While local phasing information becomes more and more important in pathogenicity studies and clinical diagnosis (*Cheng et al., 2018*; *Wu et al., 2018*), we provided an example of phasing using short reads, ABO blood type detection, in the end, to illustrate how short-read phasing can help real medical applications.

## METHODS

HAHap is a read-based haplotyping algorithm that adopts hierarchical assembly to progressively build haplotypes, starting from high-confident pairs and working toward low-confident pairs (Fig. 1). It takes predicated heterozygous variants from a variant caller as the input. We describe the concept of hierarchical assembly in Sections "Haplotype-informative reads and variant blocks" and "Hierarchical assembly"; the design of probabilistic confidence scores (CSs) in "Multinomial distribution metric"; and the local MEC search in "Local phasing using MEC" and "Examples of choices between heuristic and local MEC-based search".

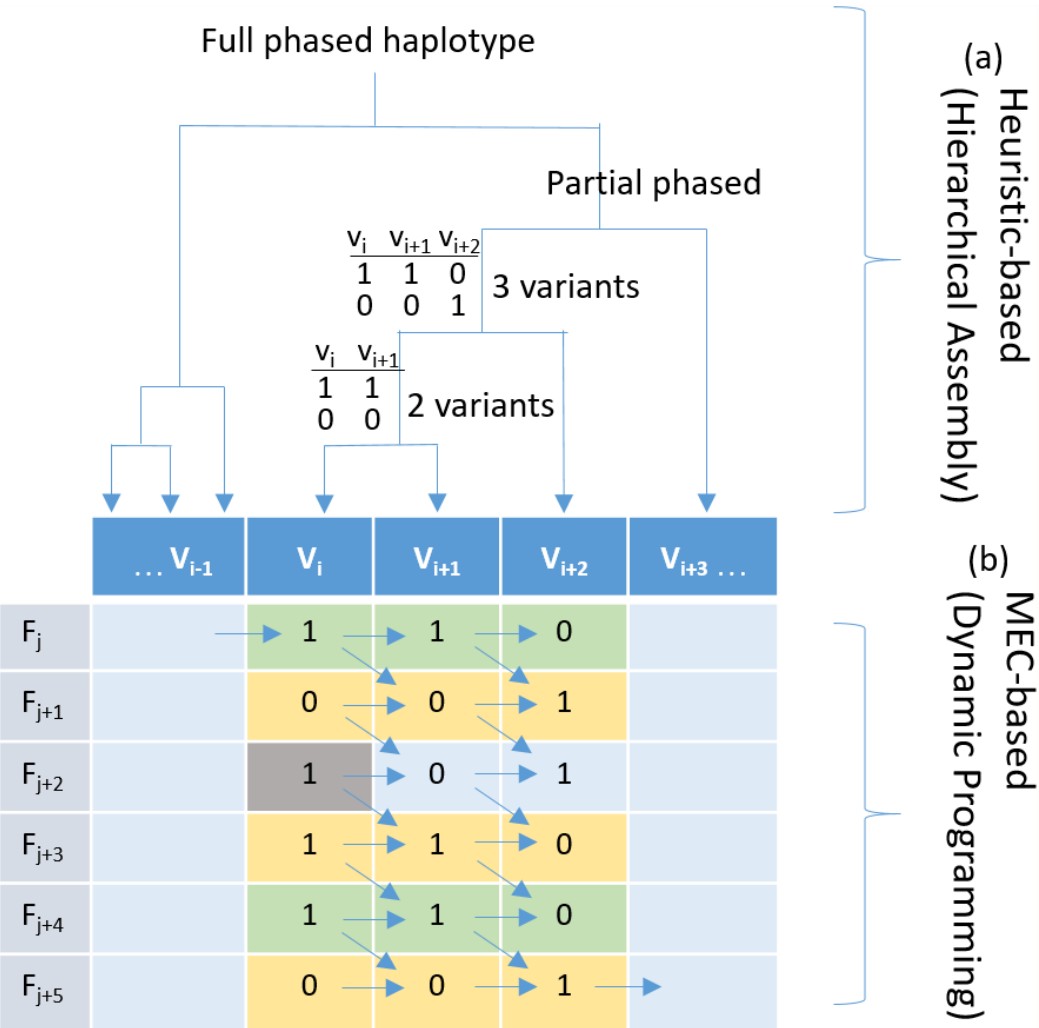

**Figure 1 Haplotyping using hierarchical assembly vs. dynamic programming.** (A) Indicates the proposed heuristic-based process using hierarchical assembly, and (B) shows the MEC-based search process using dynamic programming. $F_j$ is a fragment (read pair), and $V_i$ is a heterozygous variant. Given that the gray cell is a sequencing error, the proposed hierarchical approach will not be affected when phasing other positions, but the MEC approach will consistently consider it during the whole process, and this might damage the solution if too many noises exist.

## Haplotype-informative reads and variant blocks

A locus with the same alleles is homozygous, and a locus with more than one allele is heterozygous. In the phasing problem, we only consider heterozygous variants. The reads spanning multiple heterozygous variants are used in assembling haplotypes. The homozygous variants, which cannot provide information to extend haplotypes, are ignored. In the rest of the article, "variant" is short for "heterozygous variant". Two variants are connected if they are spanned by a read. The connectivity of variant pairs is transitive. For example, suppose there are three variants A, B and C, but no read spans A and C. However, there are reads spanning variants A and B and other reads spanning variants B and C. In this case, A and C can be joined through B, and we describe each pair

of variants among these three variants connected. In other words, the reads spanning multiple variants are haplotype-informative. In this regard, the first step of the proposed method is to identify the haplotype-informative reads and determine a variant block in which all the variants inside are connected.

DNA sequencing technology produces reads or read pairs, that is, fragments of DNA. In the field of computer science, we usually define a read as a string of nucleotides (A, T, C, G). In the phasing problem, we describe it as a string of (0, 1, -) to stand for the allele of (major, minor, others). A variant block is a set of variants where each one has at least one connection with the others. The informative reads represent reads that span at least two variants.

## Hierarchical assembly

HAHap considers each pair of two loci as pieces of the puzzle and proceeds with the assembly according to the CSs. To resolve the whole haplotypes, we start from phasing small pieces and assemble them using the concept of hierarchical assembly. Each resultant variant block in Section "Haplotype-informative reads and variant blocks" is an isolated problem for hierarchical assembly.

Although the MEC method in general capably handles noise, it can still accumulate errors and thus lead to poor phasing when noise overrides the signal in the data. According to this observation, our method endeavors to reduce the influence of noise by leveraging pairs with higher scores. This is achieved by adopting a hierarchical assembly approach, which considers the pair with the highest score first and proceeds to the end. At the beginning of the assembly, the algorithm treats each variant as a vertex and calculates a CS of two vertices for guidance. The CS of a variant pair is described in Section "Multinomial distribution metric". Then, the single-linkage metric is adopted to evaluate the score between two clusters. The term "cluster" represents a group of variants and is used in the following discussion. The algorithm continues to unify clusters until all vertices are unified. In total, a block with $n$ variants inside needs $n-1$ merging to group all variants into one unit. This process of assembly could be finished in linear time.

The score $S(X, Y)$ between two clusters $X$ and $Y$ is defined below, where $X$ and $Y$ are any two sets of variants as clusters. The CS takes two variants, $x$ and $y$, as arguments. It ranges from negative infinity to zero. A score closer to zero indicates more confidence.

$$S(X, Y) = \max_{x \in X, y \in Y} \text{CS}(x, y) \qquad \text{(single-linkage)}$$

HAHap always trusts the heterozygous variants reported by the variant caller. In order words, it only considers the major and alternative alleles detected in variant calling and treats the variants as heterozygous. This is the so-called heterozygous assumption. For example, if the variant caller determines $V_1$ and $V_2$ as two heterozygous loci, and $V_1$ is observed with C/T (0/1) and $V_2$ with G/C (0/1), the heterozygous assumption tells us that the haplotype candidate (C–G, C–C) is not considered since (C–G, C–C) turns $V_1$ homozygous. Only two solutions are preferred; one is (C–G (0–0), T–C (1–1)) and the other is (C–C (0–1), T–G (1–0)).
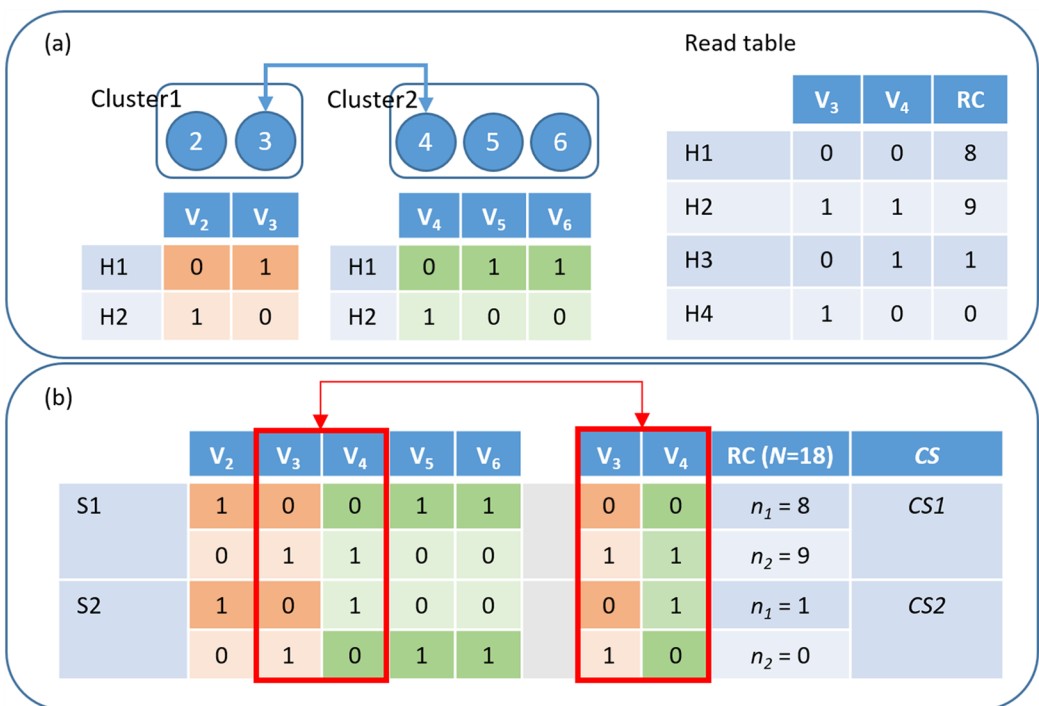

**Figure 2 HAHap uses the proposed CS score to guide the haplotyping process.** (A) Merging Cluster1 and Cluster2 by variants $V_3$ and $V_4$: There are two variants in cluster1 and three in cluster2. HAHap chooses the pair with the highest CS score to lead the process (the 3–4 pair in this example). HAHap records the read counts (RC) of four possible combinations into the read table; (B) Calculating the scores for the two preferred solutions (S1 and S2) for variants $V_3$ and $V_4$: We used the notation (0–0, 1–1) to denote the first prediction (S1), which is (Cluster1_H1-Cluster2_H1, Cluster1_H2-Cluster2_H2). Similarly, we used the notation (0–1, 1–0) to denote the second prediction (S2), which is (Cluster1_H1-Cluster2_H2, Cluster1_H2-Cluster2_H1). In the end, the one with the higher score (CS1) will be chosen as the prediction.

After explaining the rules of the hierarchical assembly, this paragraph introduces a way to resolve one large haplotype from two small phased haplotypes. Our method takes the reads spanning the variant pair of the highest score between two clusters to combine the sub-haplotypes that have been phased in previous steps. Among two preferred solutions, the first solution's alleles for the pair are (0–0, 1–1) and those of the second solution are (0–1, 1–0). We calculated the CS, CS1 and CS2, for the first and second solutions, respectively, and used the read information of the higher CS to infer a larger haplotype (Fig. 2). The hierarchical assembly continually executes until all possible pairs are considered. However, in a heuristic method, it is possible to make a wrong decision on merging with a low CS. Whenever it is likely to happen, HAHap introduces a local MEC search to overcome this problem, which will be explained in Section "Local phasing using MEC".

## Multinomial distribution metric

In this section, we introduce the scoring metric that measures the confidence between two variants. The CS is based on the multinomial distribution. Each pair of the two connected variants is treated as an instance for scoring. When appraising the confidence

level of variants $x$ and $y$, we count the total reads that span both $x$ and $y$, and denote this number as $N$. Then we consider two preferred solutions separately. First, we define the observed read count of one haplotype as $n_1$ and the observed read count of the other haplotype as $n_2$. The sum of $n_1$ and $n_2$ is the total observations coming from the solution; therefore, we expect this number to be as close to $N$ as possible. However, the errors from sequencing or alignment bring in unexpected reads that do not fit in any two haplotypes. To handle these potential errors, we introduce the third number ($n_3$) to take care of the unexpected reads. Let $P_1$ and $P_2$ be the likelihoods of the two haplotypes, and $P_3$ be the likelihood of the unexpected observations. We define the multinomial score, MS, as the $\log_2$ value of the possibility that the solution satisfies the multinomial distribution when given the observed reads. A normalizing factor, $F$, is used to normalize the effect caused by $N$. Accordingly, the normalized score, NS, is defined as the value obtained by subtracting $F$ from MS, and the symbol $C$ in formula means the combination (a selection of items from a collection) in mathematics.

$$N = \sum_{1}^{3} n_i, \sum_{1}^{3} P_i = 1, 0 \leq P_i \leq 1 \tag{1}$$

$$\text{MS} = \log_2(C_{n_1}^{N} C_{n_2}^{N-n_1} P_1^{n_1} P_2^{n_2} P_3^{n_3}) \tag{2}$$

$$F = \log_2(C_{N/2}^{N} C_{N/2}^{N} P_1^{N/2} P_2^{N/2}) \tag{3}$$

$$\text{NS} = \text{MS} - F \tag{4}$$

Here, we need to consider the effect of the sequencing coverage. Let the term "max coverage" stand for the largest read count among all pairs in this block and the term "local coverage" stand for the read count of the two variants of interest. We define the ratio of local coverage and maximum coverage as $c$, and use it to adjust the inflection point of the sigmoid function. In the end, we define CS as shown below, which is actually an adjusted value of NS.

$$c = \frac{\text{Local coverage}}{\text{max coverage}} \tag{5}$$

$$\text{sigmoid}(c) = \frac{1}{1 + e^{-(c-0.5)}} \tag{6}$$

$$\text{CS} = \log_2(\text{sigmoid}(c) \times \text{NS}) \tag{7}$$

The meaning of CS is as follows. In diploid organisms, ideally, reads should only come from the two haplotypes, and the ratio of the observed read counts from the two haplotypes should be close to 1:1. However, in real cases, sequencing and alignment errors cause the cis-allelic appearance to stray from equality. Based on this assumption, HAHap uses a multinomial distribution to estimate how likely the sequencing reads are sampled following this distribution. As a result, this score evaluates the confidence with which the reads on this pair follow the multinomial assumption. We note that there are two preferred solutions for each pair of the two variants. We choose the higher one as the CS for this pair.

The metric takes three factors to eliminate possible biases caused by different coverages. First, the multinomial score is divided by the normalized score, which represents the perfect non-skewed case of the multinomial distribution. Second, we use a sigmoid function to capture the difference between a good region and a bad region. Third, we use the parameter $c$ to adjust the inflection point of the sigmoid function to keep the score with coverage at $c/2$ unchanged. The method takes the CS to prioritize all pairs of variants and uses it to pilot the hierarchical assembly. The distance between two variants is another important clue. We assume that two closer variants are more reliable. Among pairs with the same score, the method chooses the closer one as the next assembly step.

## Local phasing using MEC

The assembly process reduces the search space dramatically. However, it has the drawback that false inferences could cause serious consequences. Therefore, under certain conditions, we perform a local MEC-based search. When merging two clusters, it is possible that the variants from the two clusters are interleaved. A junction is the boundary between two adjacent variants from different clusters. The method adopts a voting mechanism in which every read spanning the junctions is recruited to evaluate the two preferred solutions. The penalty is defined as how many corrections have to be made on reads in order to be consist with the solution. In the end, the solution with a lower penalty will be chosen. The local MEC process checks potential corrections for every read in a block. This step is time-consuming, but fortunately it only executes in local regions in HAHap. Next, we explain when the local MEC process will be invoked in the following two sub-sections.

### Embedded merging

We defined embedded merging as the case that merges two clusters with multiple junctions in between adjacent variants. In this situation, HAHap applies a local MEC-based search instead. One reason is that this kind of situation involves numerous phasing decisions, which means the algorithm should proceed more carefully. Normally, closer variants tend to have a higher CS and are usually merged earlier. In this regard, multiple junctions often happen in the area with erroneous information. In the local MEC-based search, all reads spanning the junctions are responsible for inferring haplotypes. We defined the minimum junction number as the threshold for triggering a local MEC-based search. For example, when the number of minimum junctions is set to three, the merging with three junctions or more will invoke a local MEC-based search.

There are only two potential solutions for a particular MEC-based search (mating the two sub-haplotypes from each cluster, respectively). The search only inspects all reads covering the specified region. On average, the time complexity of the local MEC-based search is proportional to the read coverage in this area.

### Only ambiguous variant pairs remain

We declare singleton pairs or low-coverage pairs as ambiguous variant pairs. A singleton pair means all reads on this pair only support one haplotype in both preferred solutions. Low-coverage pairs indicate that the count of reads spanning this pair is below the

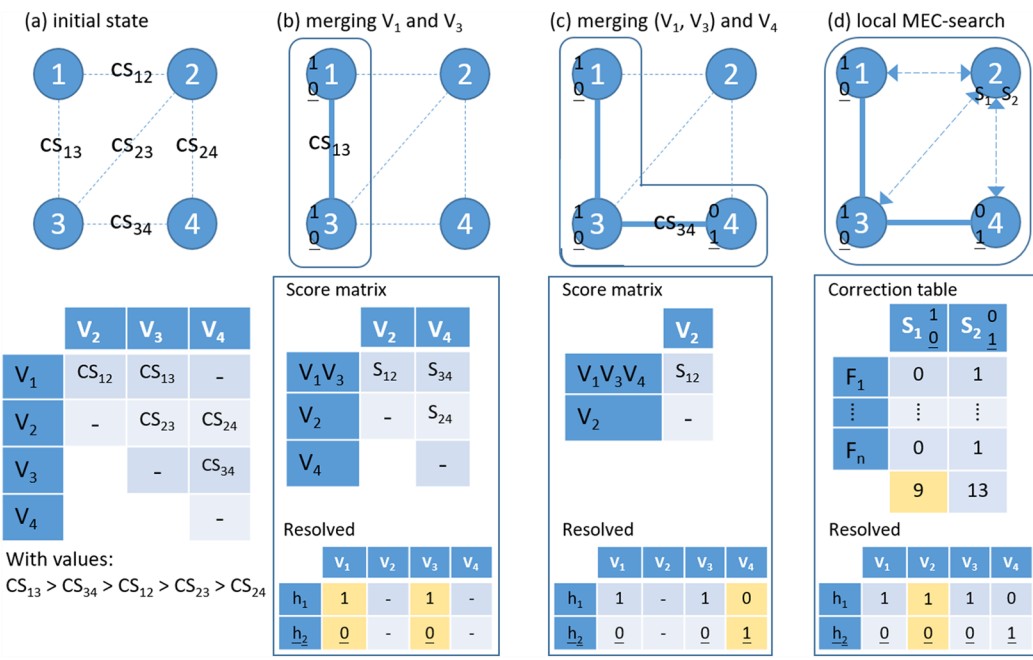

**Figure 3 An example of the proposed hierarchical assembly.** $F_j$ is a fragment (read pair), $V_i$ is a heterozygous variant and CS for confidence score. (A) Initial state; (B) Merging variants 1 and 3 and updating the score matrix by single-linkage; (C) Merging cluster (1–3) and variant 4; (D) Local MEC-search: calculating the number of error corrections for the two preferred solutions. For each sub-figures, the lower panel shows the resolved haplotypes.

threshold we defined. By default, HAHap chooses the median among all observations as the threshold. In both cases, the CSs usually fail to guide the assembly correctly. Based on the definition of CSs, the score of a singleton is quite small due to the absence of one haplotype. Similarly, low-coverage pairs provide unreliable CSs. Even though the CSs of singleton pairs are extremely low, they are observed to provide correct phasing information with only one haplotype observation. In this regard, when only ambiguous pair remains, we use singleton pairs earlier than low-coverage pairs during assembly.

Sometimes, there may be too many ambiguous pairs in a single block. In this situation, the method would be slow. To faster the method, we combine the embedded mergence with this rule. When encountering ambiguous pairs and the most strict embedded mergence case (minimum junction number is two), we perform a local MEC-based search.

## Examples of choices between heuristic and local MEC-based search

In a single nucleotide polymorphism (SNP)-partitioning graph, the vertex stands for a variant and an edge links two connected variants. Figure 3 shows a toy demonstration of phasing, which involves hierarchical assembly in the first two steps and a local MEC-based search in the last step. In this process, the weights on edges are the CS scores and the edge with the highest CS in this step is used connect the next pair of variants or variant clusters. In summary, HAHap designs a metric for conducting hierarchical assembly heuristically and switches to MEC-based searches in certain situations.

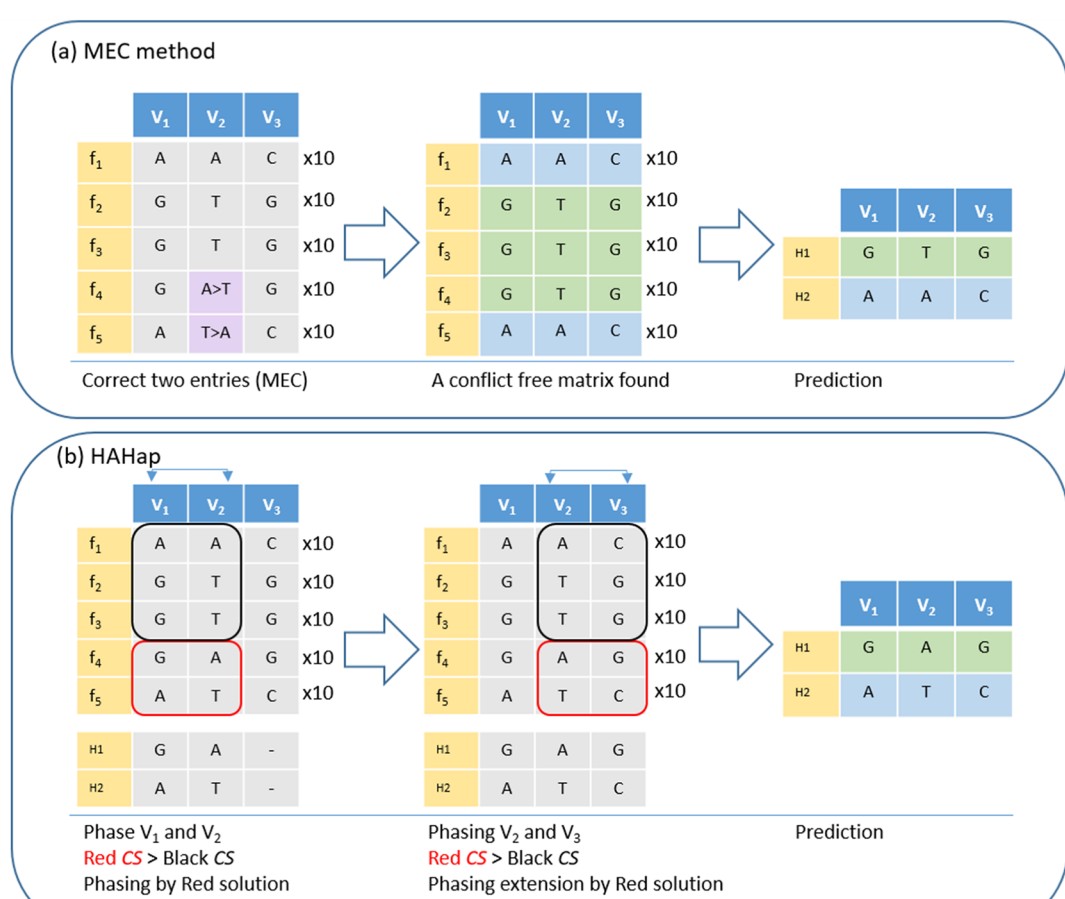

**Figure 4** **An example to demonstrate the difference between HAHap and MEC-based methods.** This example contains three variants spanned by 50 reads, including five types of reads ($f_1$–$f_5$). (A) The MEC method finds a solution with 20 corrections (purple cells) to produce a conflict-free matrix. (B) HAHap selects to phase $V_1$ and $V_2$ first according to the proposed CS scores. There are two preferred solutions: red and black rectangles. Because the sampling ratio is close to 1:1 in the red solution, the CS of the red solution is higher than CS of the black solution. Therefore, it phases the segment as (G–A, A–T). Next, HAHap extends the phasing process to $V_3$ by the same rule. In the end, two methods predict different haplotypes. The MEC method assumes reads in categories 4 and 5 are errors, but HAHap suggests reads in categories 1, 2 and 3 are incorrect.

Here, we further used an example to demonstrate that HAHap performs better than MEC-based methods in certain situations. In Fig. 4, five types of reads span a block containing three variants. The example was transformed as a SNP-variant matrix for the phasing problem. In Fig. 4A, we described how a MEC method proceeds to find the solution. In Fig. 4B, we showed that HAHap determines the phase between $V_1$ and $V_2$ first, and then extended the haplotype to $V_3$. In the end, two methods predicted different haplotypes. The MEC method assumed that the read type 4 and read type 5 are incorrect and flipped two entries to make the matrix with no conflicts in the resultant haplotypes. On the other hand, HAHap suggested read types 1, 2 and 3 are incorrect and predicted the result based on read type 4 and read type 5. In other words, the MEC-method considers the exact number of errors, while HAHap cares more about the ratio of the observed read counts from the two haplotypes. We managed

to validate which method is better in both real data and simulated data in the next section.

## EXPERIMENTAL RESULTS

We evaluated real and simulated datasets. WhatsHap and HapCut2 are two representative MEC-based tools for phasing problems (*Chaisson et al., 2018*; *Sedlazeck et al., 2018*). We compared HAHap with MEC-based tools to reveal that the proposed idea to reduce the search space and remove noise is beneficial to phasing. In HAHap, the likelihood of $P_1$ and $P_2$ were 0.49, and the minimum number of junctions for triggering MEC-local search was three as default. The other two programs were executed using default parameters.

### Evaluation measurement

Here, we introduce two concepts, "variant needing to be phased" and "phased variant". We define a variant needing to be phased as all variants except the left-most variant in its block. For example, when a block contains seven variants, the first variant does not need to be phased, and only six phasing decisions remain. On the other hand, the term "phased variants" is how many variants are indeed phased by a tool. If a locus is too difficult to phase, the algorithm could leave it as undecided. We use *number_of_variants* to stand for the first concept ("variant needing to be phased") and *number_phased* for the second one ("phased variant"). For convenience, we used the term "case" to refer a variant block in the phasing problem. This study used two measurements, phasing error rate and perfect ratio, for evaluating the accuracy of the phasing algorithms. Additionally, we incorporated a measurement, quality adjusted N50 (QAN50), to evaluate the average effective length of the properly phased blocks.

For each block, the first predicted haplotype is always a mosaic of the two true haplotypes, and the second predicted haplotype is exactly the complement of the first one, due to only considering heterozygous sites. In this regard, the switch error is defined as the case where a swap event between two haplotypes happens. However, when two switch errors are adjacent, it is treated as one flip error instead. We used *flip_error* and *switch_error* to, respectively, represent the total number of flip errors and switch errors in a block. The **phasing error rate** is defined as the sum of the switch and flip errors divided by the number of phased variants.

$$\text{Phasing error rate} = \frac{flip\_error + switch\_error}{number\_of\_phased} \qquad (8)$$

The **perfect ratio** is the ratio of the error-free phasing cases over the total phasing cases. We considered it as the true positive rate in this study:

$$\text{Perfect ratio} = \frac{number\_of\_error\_free\_case}{number\_of\_case} \qquad (9)$$

Additionally, we used QAN50 to evaluate the completeness and quality of the predicated haplotype. This measure is calculated as follows: (1) breaking each haplotype

block into the longest possible sub-blocks where no switch error inside; (2) calculating the distance from the first phased variant to the last phased variant for each sub-block; (3) multiplying the distance of each sub-block by the proportion of phased variants inside the sub-block; and (4) calculating the N50 of the distance set revised in step (3).

## Evaluation using real data

The GIAB hosted by the National Institute of Standards and Technology has characterized seven individuals using 11 different technologies (*Zook et al., 2014*). This study focused on the Ashkenazim trio, consisting of three related individuals: NA24143 (mother), NA24119 (father) and NA24385 (son). The NovoAlign BAM files of $2 \times 250$ paired-end reads with coverage around $40\times$ to $50\times$ produced by Illumina HiSeq were downloaded from GIAB and used as input for phasing. GIAB also provides the haplotypes of the Ashkenazim trio predicted by $10\times$ Genomics. The $10\times$ GemCode Technology creates a unique reagent delivery system that partitions long DNA molecules (including >100 kb) and prepares sequencing libraries in parallel such that all fragments produced within a partition share a common barcode. By this barcoding technique and combined with a proprietary data analysis tool called Long Ranger software (v2.1), the prediction could be more reliable and large-scale than other experimental phasing. This study adopted the $10\times$ Genomics prediction as the answer for haplotyping. This study considered all bi-allelic SNPs on chromosomes 1–22, resulting in 5,342,998 SNPs in the mother, 5,220,679 in the father and 4,931,224 in the son. We only considered heterozygous variants, resulting in 3,465,217 heterozygous variants in the mother, 3,406,189 in the father and 3,108,937 in the son. Following the standard pipeline, phasing after variant calling, we used GATK HaplotypeCaller (v3.6) as the variant caller to forecast the variants, which identified the heterozygous loci for phasing tools. We took the intersection of the predictions from $10\times$ Genomics and the variant calling results from GATK as the ground truth to make sure the interested variants are indeed covered by the raw reads. Finally, in the answer set of real data experiment, there are 2,240,300 heterozygous variants in the mother, 2,174,189 in the father and 2,225,891 in the son, which are considered detectable in the Illumina BAM files. Each tool has unique features for identifying variant blocks, for example, different mapping quality cutoffs and re-alignment procedures, so that each tool classifies distinct variant blocks with different numbers of variants inside. To have a fair comparison, we used all reads without filtering to recognize the original distribution of the block sizes (defined as the number of heterozygous variants in a block). The blocks with sizes less than 20 account for 99% (Table S1) of the cases. Since the challenge of phasing is on the larger blocks, we emphasized our investigation of cases containing variants more than 20.

We discussed the perfect ratios first. Among the blocks with sizes more than 20, we identified 12,784 cases in the BAM files before read filtering and took this number as the original case number. In this condition, HAHap successfully identified more error-free phasing cases (12,191 perfect cases; true positives) than WhatsHap (11,712 cases) and HapCut2 (12,125 cases). Table 1 and Table S1 listed the results for all cases, and Table 2 highlights the results for case sizes larger than 20. To investigate the

**Table 1 Evaluation results of all the cases in real datasets.**

| Method | Perfect ratio (%) | Phasing error rate (%) |
| --- | --- | --- |
| HAHap | 99.39 (1,277,260/1,285,153) | 0.2293 (8,646/3,771,159) |
| WhatsHap | 99.15 (1,272,347/1,283,302) | 0.3169 (10,955/3,774,198) |
| HapCut2 | 99.35 (1,276,857/1,285,092) | 0.2503 (9,440/3,771,022) |

**Table 2 Evaluation results of the cases with size >20 in real datasets.**

| Method | Perfect ratio (%) | Phasing error rate (%) |
| --- | --- | --- |
| HAHap | 95.36 (12,191/12,784) | 0.2283 (1,056/462,542) |
| WhatsHap | 91.14 (11,712/12,851) | 0.5105 (2,405/471,133) |
| HapCut2 | 94.87 (12,125/12,780) | 0.2992 (1,384/462,564) |

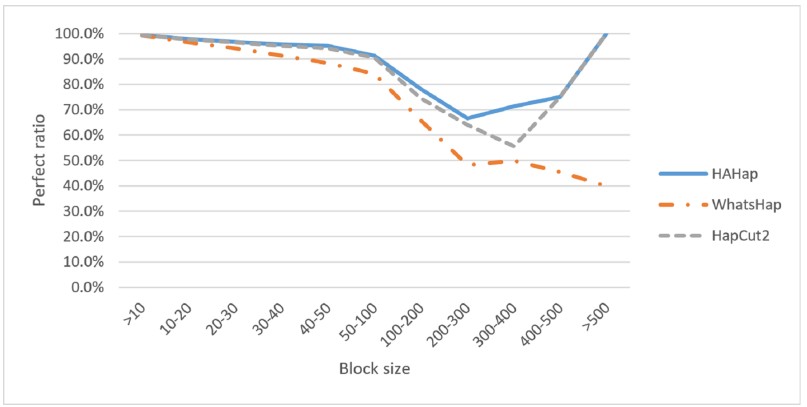

**Figure 5 Comparison on perfect ratios.**

performance relative to block sizes, the true positive rate across different block size is shown in Fig. 5. This reveals that HAHap outperformed the others in all categories of block sizes.

Next, we compared the phasing error rates between tools. HAHap performed better (0.228%) in this measurement than WhatsHap (0.511%) and HapCut2 (0.299%). Regarding the block size, HAHap consistently outperformed the others across categories. The results are shown in Fig. 6.

We used scatter plots to visualize the comparison and verified the significance of differences. In Figs. 7 and 8, we only considered the blocks with sizes larger or equal to 20. Because the three tools predicted perfectly on most of the cases, we excluded those cases in the following statistic testing. After filtering out the cases with "phasing error rate" = 0 on both tools in comparison, only 447 cases remained in the comparison of HAHap and WhatsHap and 485 cases remained in the comparison of HAHap and HapCut2. The $p$-values of Wilcoxon rank sum test are 8.969E-06 when comparing HAHap vs. WhatsHap and 0.0181 when comparing HAHap vs. HapCut2. Besides, we observed that the dots below the diagonal lines are much more than the ones above the diagonal lines in both figures. This revealed that WhatsHap and HapCut2 predicted more problematic phased variants (having higher error rates) than HAHap.

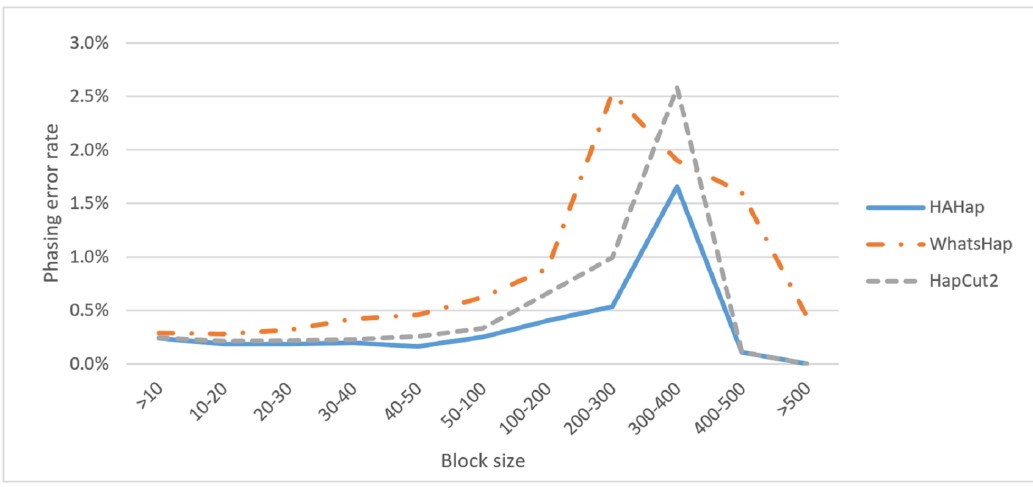

**Figure 6 Comparison on phasing error rates.**

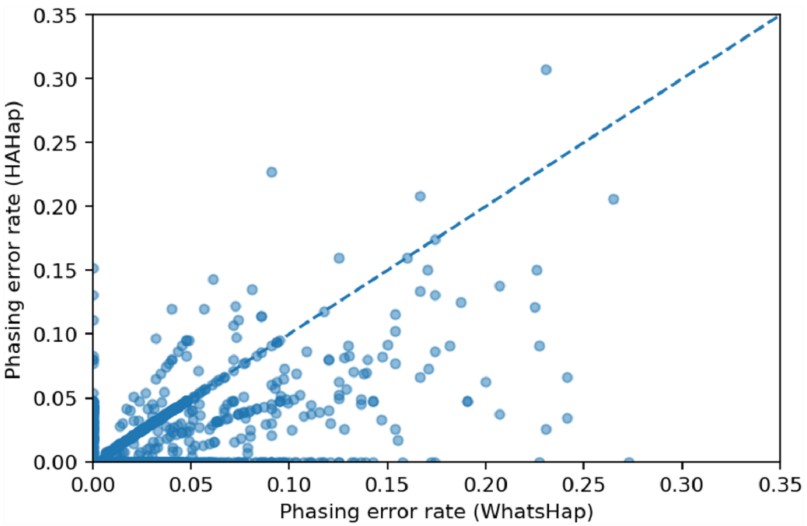

**Figure 7 Comparison of HAHap with WhatsHap in blocks with a size ≥20.**

The last measure, in terms of completeness and quality, QAN50, showed closer phasing quality between these three programs (Table 3 and Table S1). The QAN50 for blocks with a size large than 20 is 4,295 for HAHap and are 4,367 and 4,275 for WhatsHap and HapCut2, respectively. WhatsHap delivered slightly longer correct haplotypes than HAHap and HapCut2.

## Evaluation using simulated data

We conducted 50 runs of simulation with different parameter settings to investigate the performance changes under different situations. First, in each simulation, we used in-house scripts to create simulated SNPs. A genome sequence on chr22, from 16,070,000 to 16,790,000, were chosen as the experimental region, and we created 10 variant

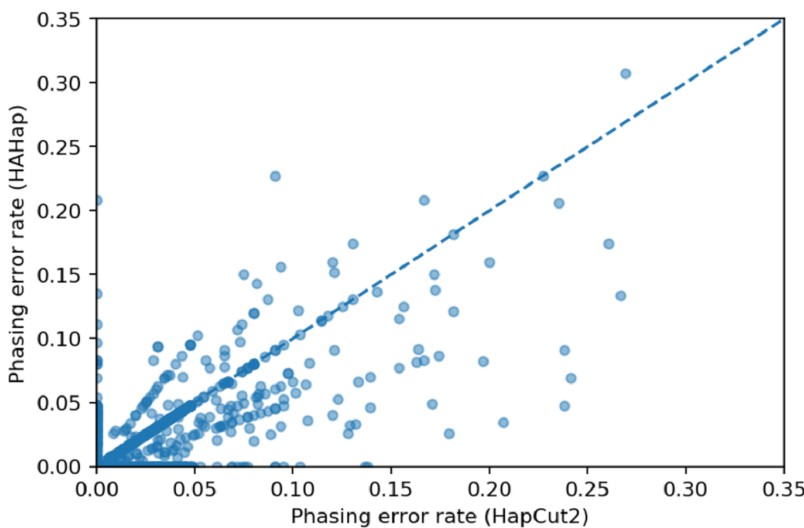

**Figure 8 Comparison of HAHap with HapCut2 in blocks with a size ≥20.**

**Table 3 Evaluation of QAN50 on real datasets.**

| Method | All cases | The cases with size >20 |
| --- | --- | --- |
| HAHap | 858.86 | 4,295 |
| WhatsHap | 864 | 4,367 |
| HapCut2 | 858 | 4,275 |

blocks for each of the five block sizes (containing 30, 50, 100, 200 and 500 variants) on either one of the haplotypes. Second, based on the artificial haplotypes, we simulated reads in each of the nine conditions to create a unique experiment. The nine conditions includes three levels of sequencing error rates (0.002, 0.01 and 0.03) and three levels of read coverages (20, 30 and 40). In total, the evaluation included 450 experiments, that is, 50 replicates for each of the nine sequencing conditions.

Simulated reads were produced by wgsim. The read length of a paired-end read is 250 bps, and the fragment length follows a normal distribution with a mean of 850 and a standard deviation of 50. We adopted BWA-MEM as the mapper and all the chromosomes of hg19 as the reference, and then took the mapped files as the input for phasing.

Here, we discussed the phasing error rates first. Among the nine sequencing conditions, shown in Fig. 9 and Table S2, HAHap performed better than WhatsHap in all conditions and outperformed HapCut2 in most conditions (better in seven conditions and worse in two). The results are consistent with the results in the real data. On the other hand, when evaluating the performance according to block sizes, as shown in Fig. 10 and Table S2, HAHap outperformed the other two tools in most of the block sizes (except the block size 200–500 when compared with HapCut2). Second, the perfect ratios of these three tools are pretty close to each other (all above 98%), and the detailed results are shown in Table S2. The number of blocks is small in the

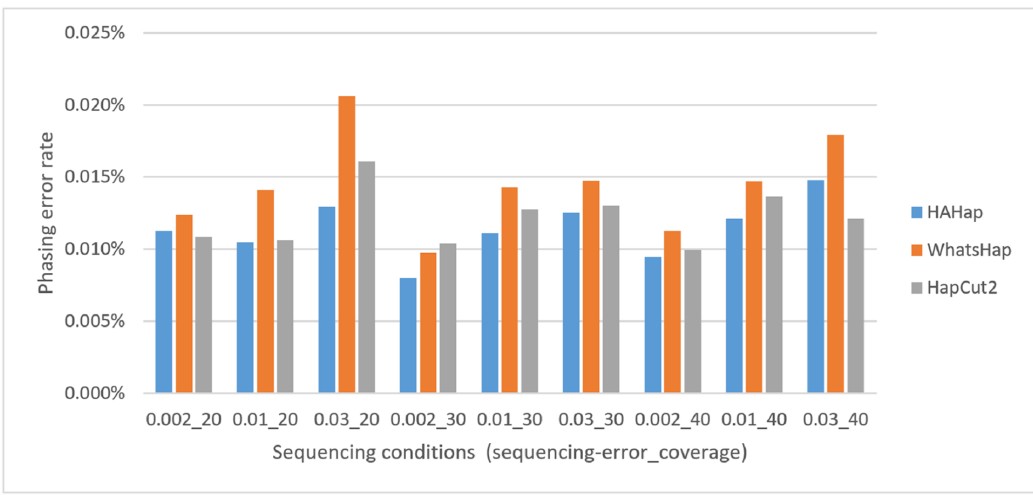

**Figure 9 Comparison on phasing error rates using the simulated data under different conditions (condition notation: "sequencing error rate"_"read coverage").**

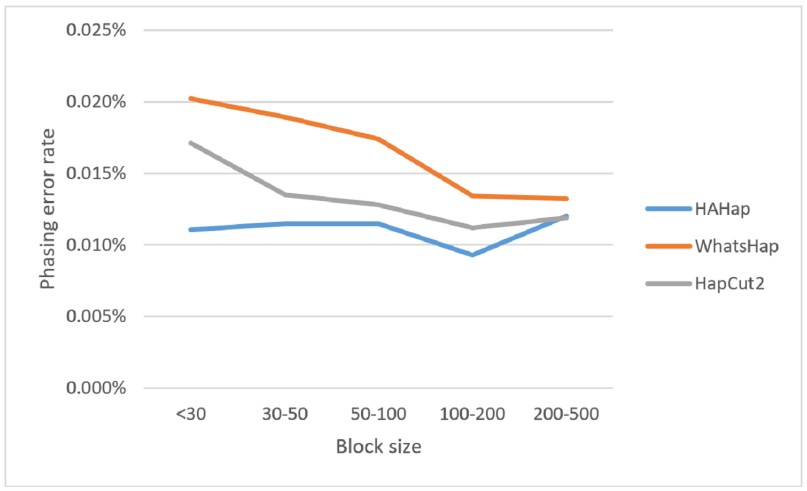

**Figure 10 Comparison on phasing error rates using the simulated data for different block sizes.**

simulation, causing the QAN50 to be less meaningful. The details of QAN50 comparison were provided in Table S2.

## Evaluation HAHap on sequencing skewness

We investigated the performance of HAHap in one more sequencing condition, sequencing skewness. Skewness means the sequencing read unbalance on two haplotypes. After integrating five levels of skewness into the previous simulation design, where nine conditions were included when considering two factors, we had 45 conditions in this examination. We did 25 times of simulation and inspected the effect of the three factors, respectively. Results of evaluating 45 distinct configurations were exhibited in Table 4 and Table S3.

**Table 4 Evaluation of 45 distinct configurations.**

| Coverage | S-e | S-skew | P-e (%%) | S-skew | P-e (%%) |
|---|---|---|---|---|---|
| 20 | 0.002 | 50/50 | 1.2798 | 20/80 | 1.0249 |
| | | 40/60 | 1.2289 | 10/90 | 1.8478 |
| | | 30/70 | 1.6000 | | |
| | 0.01 | 50/50 | 1.2414 | 20/80 | 3.2312 |
| | | 40/60 | 1.2223 | 10/90 | 14.7022 |
| | | 30/70 | 0.8636 | | |
| | 0.03 | 50/50 | 1.2887 | 20/80 | 12.1024 |
| | | 40/60 | 2.3514 | 10/90 | 92.9313 |
| | | 30/70 | 2.6748 | | |
| 30 | 0.002 | 50/50 | 0.9466 | 20/80 | 1.4463 |
| | | 40/60 | 1.1406 | 10/90 | 1.5991 |
| | | 30/70 | 1.1039 | | |
| | 0.01 | 50/50 | 1.0248 | 20/80 | 1.7626 |
| | | 40/60 | 1.2779 | 10/90 | 12.2350 |
| | | 30/70 | 0.8631 | | |
| | 0.03 | 50/50 | 1.1988 | 20/80 | 4.1918 |
| | | 40/60 | 0.9830 | 10/90 | 62.1470 |
| | | 30/70 | 2.3778 | | |
| 40 | 0.002 | 50/50 | 0.9885 | 20/80 | 1.1197 |
| | | 40/60 | 1.4362 | 10/90 | 1.2762 |
| | | 30/70 | 0.9022 | | |
| | 0.01 | 50/50 | 1.3188 | 20/80 | 1.3169 |
| | | 40/60 | 0.6999 | 10/90 | 8.2558 |
| | | 30/70 | 0.8605 | | |
| | 0.03 | 50/50 | 1.4048 | 20/80 | 2.6704 |
| | | 40/60 | 1.4719 | 10/90 | 36.6091 |
| | | 30/70 | 1.8134 | | |

**Note:**
S-e, sequencing error; S-skew, sequencing skewness; P-e, phasing error rate (%% = 1/10,000).

Under the same conditions, higher coverage and lower sequencing error rates usually facilitate better phasing performance. Only a few exceptions exist, but the difference is tiny (e.g., the combination of "coverage = 40", "error rate = 0.002" and "skewness = 50/50" is 0.00041% worse than the combination of coverage 30 with the other factors the same).

Skewness is a factor that attracts attention because the proposed CS is based on a multinomial distribution where the likelihoods of the first two outcomes are presumed to be equal. In this regard, a skewness that violates the assumption would confuse the assembly process. Table 4 reveals that with a skewness of 10/90, HAHap performed much worse than 50/50. We observe that extreme skewness leads to more singletons and pairs with much lower scores, which eventually damages the method. Fortunately, HAHap still performs well with skewness of 30/70 (with a phasing error rate close to that of 50/50), and 10/90 is an extreme case that rarely happens in real sequencing. In summary, we recommend not using HAHap under conditions of extreme skewness

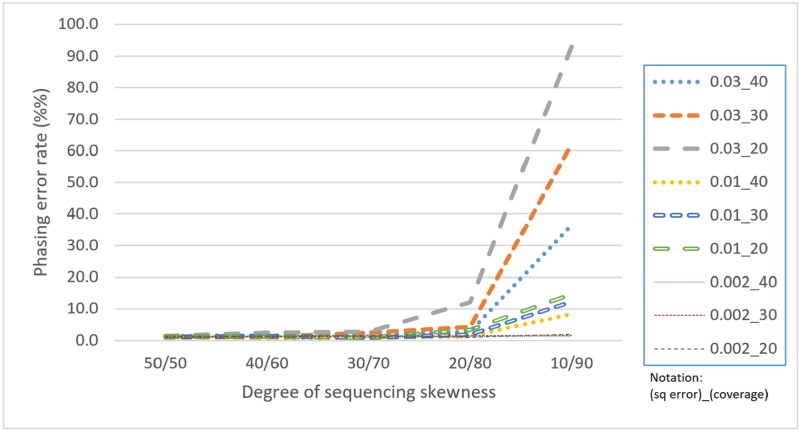

**Figure 11 Comparison of phasing error rates under different settings of skewness (Condition notation: "sequencing error rate"_"read coverage").**

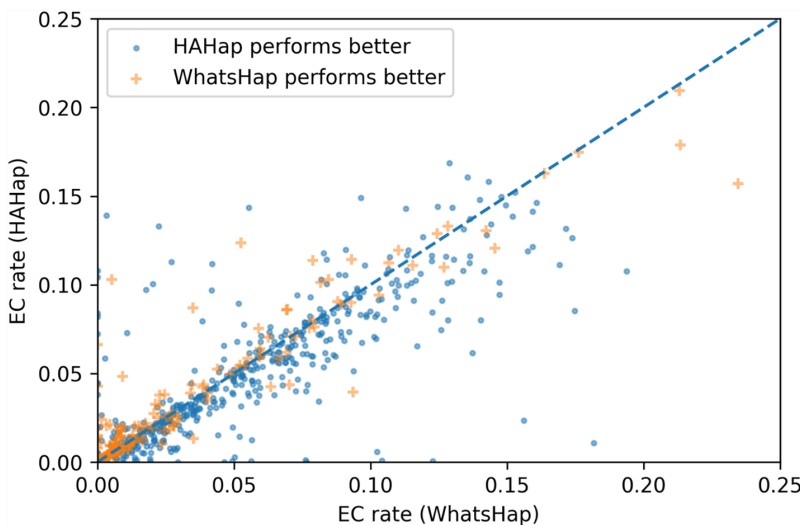

**Figure 12 Error correction (EC) rates of HAHap vs. that of WhatsHap on 700 cases of block size ≥ 20 where two tools have different predictions.**

(Fig. 11) or when the coverage is under 20. Generally speaking, HAHap is a reliable phasing method under various conditions.

## Comparison of number of error corrections

To evaluate the performance in terms of the number of error corrections (ECs) as in MEC-based approaches, we compared the number of ECs between HAHap and the other methods using real data. The EC rate is calculated as the percentage of ECs over the total characters covering reads. We drew each case on a scatter plot, where the $y$-axis stands for the EC rate of HAHap, and the $x$-axis for the tool to be compared. Both Figs. 11 and 12 use distinct symbols to label which method achieved better phasing error rates. In comparison between HAHap and WhatsHap (Fig. 12), we investigated 700 cases for which two tools have different predictions. HAHap outperformed

**Table 5 Comparison of weighted ECs rate with WhatsHap.**

| | HAHap performs better (547 cases) | WhatsHap performs better (153 cases) |
|---|---|---|
| *HAHap* | 0.0628 | 0.0460 |
| *WhatsHap* | 0.0633 | 0.0451 |

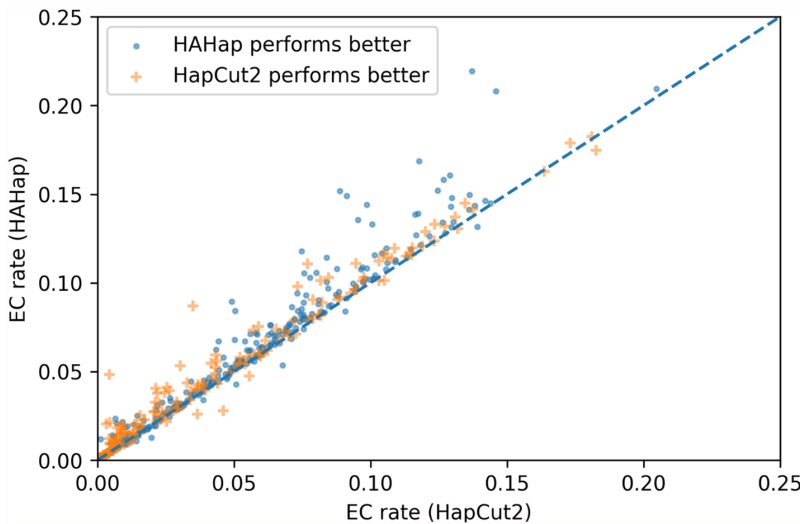

**Figure 13 Error correction (EC) rates of HAHap vs. that of HapCut2 on 475 cases of block size ≥20 where two tools have different predictions.**

**Table 6 Comparison of weighted ECs rate with HapCut2.**

| | HAHap performs better (300 cases) | HapCut2 performs better (175 cases) |
|---|---|---|
| *HAHap* | 0.0761 | 0.0636 |
| *HapCut2* | 0.0680 | 0.0592 |

WhatsHap in 547 cases, where the weighted average of the EC rates of HAHap is 0.0628, and that of WhatsHap is 0.0633. In contrast, WhatsHap surpassed HAHap in 153 cases, where the weighted average EC rate of HAHap is 0.0460, and that of WhatsHap is 0.0451 (Table 5). In the comparison to HapCut2 (Fig. 13), HAHap outperformed HapCut2 in 300 cases, where the weighted average EC rate is 0.0761, and that of HapCut2 is 0.0680. HapCut2 surpassed HAHap in 175 cases, where the weighted average EC rate of HAHap is 0.0636, and that of HapCut2 is 0.0592 (Table 6).

In conclusion, HAHap achieves a competitive level of ECs compared to the other methods we evaluated, even though it adopted a different approach rather than minimizing the number of ECs. Although HapCut2 performed better regarding ECs, HAHap was superior in most cases, which indicates that a lower rate of EC does not guarantee better performance in terms of phasing error rates.

## Application on ABO blood type detection

We cooperated with the Taipei Blood Center of the Taiwan Blood Services Foundation to demonstrate the possibility of using genotyping and haplotyping to assist traditional

| Table 7 Comparison of running time. | | | |
|---|---|---|---|
| App | NA24385 | NA24149 | NA24143 |
| HAHap | 160m13s | 182m54s | 195m58s |
| WhatsHap | 463m58s | 481m50s | 531m20s |
| HapCut2 | 52m50s | 57m50s | 67m48s |

blood typing with serology. The ABO blood group system is used to denote the presence of the A and B antigens on erythrocytes. The ABO blood group can be characterized into four main types: (1) only A antigen presents, (2) only B antigen present, (3) both presents and (4) both not present. These four types are classified as group A, group B, group AB and group O. However, the blood subtypes are much more complicated. Until today, there were more than 300 of distinct ABO blood subtypes discovered. In human blood transfusions, a mismatch on the ABO blood type between the donor and the recipient could cause a serious adverse reaction and may lead to fatality. Traditionally, the serology, through blood testing, is the standard method to determine the ABO blood group, but it has limitations with sensitivity, and manual testing may be biased to each personnel's judgment. We chose 12 samples from a previous study (*Wu et al., 2018*) to demonstrate how variant phasing can help. Five of the 12 samples belonged to the normal ABO type and served as controls here, and the remaining seven samples are unknown blood types. For these unknown samples, medical technologists found a discrepancy using ABO serology and suspected them to be ABO subtypes. Experts in the Taipei Blood Center manually determined the subtypes for these seven samples. With HAHap, we were able to find the variations, defining ABO subtypes, and to determine the cis/trans association to the A/B/O alleles without the help of the experts. Through the well-studied relations between ABO subtypes and the ABO genomic sequences, accurate haplotypes provided great enhancement when compared to the original blood type testing accuracy and therefore improved the safety of blood transfusion. Among these 12 samples, the subtypes can be determined by using the haplotypes of the 15 variants plus one additional variant for the specific subtype. These variants are located at exon 6 and exon 7 of the ABO gene. In the end, HAHap predicted both normal and subtype samples correctly. We provided the predictions in Table S4. Although only a small number of variants were included in determining the ABO subtypes, these examples revealed the value of conducting haplotyping using short reads in many precision medicine applications in the future.

## Running time

We measured the running time of the three tools on the NovoAlign-generated BAM files of the Ashkenazim trio, which were used in the real data evaluation. All experiments were performed on a 48-core machine with Intel Xeon E5-2683 CPUs and 384 GB of memory. HAHap took about 3 h to complete whole-genome phasing, which was slower than HapCut2, but much faster than WhatsHap (Table 7).

## DISCUSSION

This study presents a read-based heuristic method using hierarchical assembly to build haplotypes. HAHap uses a CS to rank the pairs of heterozygous variants. The idea is that, for diploid organisms, all reads come from two chromosomes, and the counts of observations on the two haplotypes tend to be close to each other. The CS follows the diploid assumption using a multinomial distribution and takes a relatively low likelihood outcome to represent sequencing and mapping errors. Unlike MEC-based methods that try to consider all observations, including noise, HAHap attempts to exclude the influence of the noise via hierarchical assembly.

We compared HAHap with two state-of-the-art tools: an exact MEC-based method, WhatsHap, and a heuristic MEC-based method, HapCut2. The results demonstrated that our method achieves better accuracy in all categories of variant blocks on the real dataset of Illumina sequencing reads and simulated data. The results supported the idea that correctly removing noise has the potential to deliver better results. On a simulated dataset, we tested HAHap under different sequencing conditions. We considered whether the CSs based on the diploid assumption were stable in some extreme cases. The results revealed that our method handled high sequencing errors and was stable for a read set with coverage more than $20\times$. The degree of sequencing skewness encumbered HAHap the most. However, it still performed well at skewness of 30/70. A level of skewness worse than 30/70 rarely occurs. As a result, HAHap is a practical solution in most cases.

## CONCLUSION

This study proposed a hierarchical assembly-based haplotyping method, HAHap, for short-read sequencing technologies. Both real and simulated data revealed the value of HAHap in handling haplotyping for large variant blocks. In the future, moreover, it might be possible to incorporate HAHap with some other knowledge-based phasing methods like genetic and statistic phasing to obtain higher accuracy by using valuable information of related individuals.

## ACKNOWLEDGEMENTS

Thank Dr. Chung-Tsai Su, Dr. Mei-Ju Chen and Mr. Yin-Hung Lin for contributing ideas in method development.

### Funding

This work was supported by the Ministry of Science and Technology of Taiwan (No. 105-2221-E-002-129-MY3). The funders had no role in study design, data collection and analysis, decision to publish, or preparation of the manuscript.

### Grant Disclosures

The following grant information was disclosed by the authors:
Ministry of Science and Technology of Taiwan: 105-2221-E-002-129-MY3.

## Competing Interests

The authors declare that they have no competing interests.

## Author Contributions

- Yu-Yu Lin conceived and designed the experiments, performed the experiments, analyzed the data, contributed reagents/materials/analysis tools, prepared figures and/or tables, authored or reviewed drafts of the paper, approved the final draft.
- Ping Chun Wu conceived and designed the experiments.
- Pei-Lung Chen conceived and designed the experiments.
- Yen-Jen Oyang conceived and designed the experiments.
- Chien-Yu Chen conceived and designed the experiments, authored or reviewed drafts of the paper, approved the final draft.

## Data Availability

GitHub: https://github.com/ifishlin/HAHap

1. The novoalign BAM files for the Illumina HiSeq "2x250" reads of the Ashkenazi/PGP trio (from GIAB)

Data description (README): ftp://ftp-trace.ncbi.nih.gov/giab/ftp/data/AshkenazimTrio/HG002_NA24385_son/NIST_Illumina_2x250bps/novoalign_bams/README;

Ashkenazi son. (HG002.hs37d5.2x250.bam) ftp://ftp-trace.ncbi.nih.gov/giab/ftp/data/AshkenazimTrio/HG002_NA24385_son/NIST_Illumina_2x250bps/novoalign_bams/;

Ashkenazi father. (HG003.hs37d5.2x250.bam) ftp://ftp-trace.ncbi.nih.gov/giab/ftp/data/AshkenazimTrio/HG003_NA24149_father/NIST_Illumina_2x250bps/novoalign_bams/;

Ashkenazi mother. (HG004.hs37d5.2x250.bam) ftp://ftp-trace.ncbi.nih.gov/giab/ftp/data/AshkenazimTrio/HG004_NA24143_mother/NIST_Illumina_2x250bps/novoalign_bams/.

2. The haplotypes of the Ashkenazim Trio predicted by 10xGenomic (from GIAB)

Data description (README) ftp://ftp-trace.ncbi.nih.gov/giab/ftp/data/AshkenazimTrio/analysis/10XGenomics_ChromiumGenome_LongRanger2.1_09302016/README;

Ashkenazi son. (NA24143_hg19_phased_variants.vcf) ftp://ftp-trace.ncbi.nih.gov/giab/ftp/data/AshkenazimTrio/analysis/10XGenomics_ChromiumGenome_LongRanger2.1_09302016/NA24143_hg19/;

Ashkenazi father. (NA24149_hg19_phased_variants.vcf) ftp://ftp-trace.ncbi.nih.gov/giab/ftp/data/AshkenazimTrio/analysis/10XGenomics_ChromiumGenome_LongRanger2.1_09302016/NA24149_hg19/;

Ashkenazi mother. (NA24385_hg19_phased_variants.vcf) ftp://ftp-trace.ncbi.nih.gov/giab/ftp/data/AshkenazimTrio/analysis/10XGenomics_ChromiumGenome_LongRanger2.1_09302016/NA24385_hg19/.

3. wgsim read simulators

https://github.com/lh3/wgsim.

## Supplemental Information

Supplemental information for this article can be found online at http://dx.doi.org/10.7717/peerj.5852#supplemental-information.

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
