# Peer review of "HAHap: a read-based haplotyping method using hierarchical assembly"

_PeerJ, doi:10.7717/peerj.5852_

## Round 0.1 · original submission · Major Revisions

All reviewers and myself agree that the article needs substantial additions and clarifications in order to come to an acceptable state. As we do want to keep the burden to reviewers low (and consequently, the number of iterations), please only re-submit after addressing all of the reviewers comments in a major re-write of the manuscript, accompanied by a point-by-point response to the comments.

Reviewer 1 ·

Basic reporting

no comment

Experimental design

no comment

Validity of the findings

no comment

Additional comments

I have several questions and suggestions of this paper
1. Like WhatsHap and HapCut2, the authors should implement their algorithm as a software tool for public use.
2. Can HAHap be used in organisms which do not have reference genomes?
3. Why do authors use old hg19 genome in the simulate datasets? The authors should use the latest hg38 genome if possible.
4. Why do the authors use the phasing error rate and perfect ratio as the only performance indices? Are there any other performance indices used in the existing algorithms? If yes, why not use them?
5. The authors compared their algorithm with WhatsHap and HapCut2 only using one real data. I would like to see the comparison in the simulated data as well. In that case, I would be convinced that HAAap really outperforms WhatsHap and HapCut2.

·

Basic reporting

No comment

Experimental design

The overall experimental design is clear and intuitive. The author applied the multinomial distribution to assess the confidence score between two variants, which is the fundamental of the HAHap. Since the confidence score is calculated from a likelihood-based method, I am wondering if the confidence score is affected by the sample size, (here, the sample size could be N). For example, the variant pair with larger N comes with higher confidence score. If it is so, the authors might need to discuss it.

Validity of the findings

The authors have provided the comprehensive comparisons between HAHap and the other two algorithms. All the bioinformatics algorithms/software are designed to investigate or help to investigate the biological science. I suggested that the authors might apply the HAHap to some real data and demonstrate that the HAHap is capable of identifying the well-known disease-associated haplotypes. Through this way, I believe that the applicability of the HAHap could be emphasized more to the readers.

Additional comments

1. I noticed that, in the Fig. 4 and 5, the trends were changed at the block size of 300-400. I am wondering why this happened. The authors might discuss it in the main text.
2. I suggested the authors to add boxplots in Fig. 6 and 7 to show the difference of error rates between the evaluated algorithms. In addition, the paired Wilcoxon ranksum test could be helpful to assess the significance of the differences.

·

Basic reporting

The authors propose an approach for the (hierarchical) haplotype assembly of short reads whose objective function is the minimum error correction (MEC).

While this method may show promise, it is not quite clear from the exposition, what exactly is hierarchical assembly, and Figure 1 does not seem to make this any more clear.

Moreover, the discussion is not very deep, i.e., the MEC is not the only objective function to consider, and the authors miss out on much of the related literature. Aside from this, different methods optimize the MEC in different ways, e.g., Whatshap is optimized for low-coverage long-reads (such as PacBio and ONT), and has since been augmented to consider pedigrees as an extra source of information for phasing.

Experimental design

For reasons that likely result from the problems indicated in the previous section, the experimental study seems inconsistent. Indeed, the authors do consider a trio (a pedigree of size 3) of long PacBio reads — which is the ideal input for WhatsHap, for example — however there is nothing about whether HAHap also phases trios (or just single individual, like HapCUT2 considers, for example), let alone whether it works best for long reads (one would suppose not, since in the abstract it is stated that it is for “.. constructing haplotypes using short reads ..”. Why not compare to HapCol, for example? It was shown to outperform Whatshap in certain contexts such as this. Moreover, using GATK as a ground truths seems out of date, since its ReadbackedPhasing project has been discontinued for at least a few years — one could rather use GIAB gold-standard phasings instead (available for Ashkenazim trio, as well as NA12878)

Also, the simulated study seems incomplete. Why not compare HAHap to the other methods on the simulated data, where the ground truth is known?

Validity of the findings

no comment

Additional comments

For the above reasons, I believe the paper cannot be accepted as-is. The exposition needs to be clearer, and a more consistent and comprehensive experimental setup needs to be carried out

Reviewer 4 ·

Basic reporting

The paper “A read-based haplotyping methods using hierarchical assembly” introduces a novel phasing algorithm (HAHap). It uses hierarchical assembly to construct haplotypes from short read sequencing data. The authors aim to reduce the influence of sequencing or alignment errors on the phasing procedure. The idea is to assemble short haplotype fragments based on confidence scores computed for each pair of variants connected by sequencing reads. In regions of low confidence, haplotype segments are combined based on locally applying minimum error correction (MEC).
HAHap was evaluated on real and simulated Illumina data. It was compared to WhatsHap and HapCut2, which both solve the minimum error correction problem (MEC) in order to derive haplotypes.

In general, the paper is well structured. In the introduction, the authors give a good overview of different approaches to phasing as well as strengths and weaknesses of existing methods.
Furthermore, the outline of their new algorithm is described clearly. The authors provide figures demonstrating how their algorithm works. They are generally useful and make it easier for the reader to understand the proposed method. However, it is not described how the local MEC-based search works which is applied in low confidence regions. Since this seems to be an important aspect, more details should be given.
In some parts of the paper (especially the evaluation section), the writing needs to be improved since language and notation are sometimes ambiguous, making it difficult for the reader to understand some parts of the paper (see Validity of the Findings and General Comments for examples).

The authors provide the data which was used for their evaluation experiments and their tool is publicly available.

Experimental design

As a motivation for their novel algorithm, the authors claim that the fact that MEC based approaches compute solutions in a global manner can lead to wrong haplotypes in noisy regions. This should be explained in more detail since intuitively, considering all positions together should be useful in detecting sequencing errors in the reads. At least a small example should be added to demonstrate this.

Furthermore, it is not clear which objective function the authors attempt to optimize with their method. In section 2.5, the authors claim that their algorithm is similar to solving a minimum spanning tree problem and that the solution can be interpreted as a minimum likelihood solution. This requires more explanation, since it is not clear why one is interested in the minimum and not the maximum likelihood solution.

The authors claim that HAHap successfully removes noise in the data and therefore leads to better phasing results than other approaches. However, they only evaluate their method on short reads with low sequencing error rates. In order to demonstrate that HAHap indeed better handles noise, experiments should be performed on long sequencing reads which are affected by much higher sequencing error rates and for which removing noise is essential in order to achieve good phasing results.

Validity of the findings

The authors describe how they evaluated their algorithm in detail. Besides performing experiments on simulated data, they also ran their algorithm on real data, showing that it is applicable to real world sequencing data. However, when describing their experiments, some definitions and terms are unclear, for example “phasing cases” is not properly defined. Similarly, it is not clear what the authors mean by “bins” and “bin sizes” (e.g. in line 282). This makes it difficult for the reader to understand the results presented in the paper.
Additionally, since data of a trio was used for evaluation, results should be given for each individual separately.

In general, the authors give a detailed discussion of the results and use several tables and plots to visualize the most important aspects.

Additional comments

Line 147: what is meant by “each pair of the two connected variants”?

Line 161: the notation is not clear. What is c?

In Figure 2 (b), CS1 and CS2 is used to refer to the confidence scores of the two haplotype solutions. The notation should be introduced in the text.

Line 241: formulation is confusing. “The term 'phased variant' is how many variants are phased in the group 'variant needing to be phased'” suggests that these terms refer to groups of variants, but the names indicate that single variants are considered.

Line 254: when defining the perfect ratio, it is unclear what is meant by “error free phasing cases” or “total number of cases”. What is a case?

Line 273: Why was GATK run additionally when variant positions and genotypes where provided by GIAB already as mentioned earlier? And also, it should be mentioned how many variants are in the intersection.

When trying to run HAHap on a small example region, it gave this error:
ValueError: invalid literal for int() with base 10: 'chr3'

---

## Round 0.2 · Major Revisions

One reviewer (#4) mentions a number of (small) possible improvements, concerning further clarifying the method, and more importantly, getting the software to run without a KeyError.

The concerns raised by the other reviewer (#3) are more substantial, and, after considering them, I agree. Essentially, given the scope of the tool, it is currently not compared against the appropriate competitors, and also, the field has moved away from haplotyping by short reads. Therefore the value of the contribution is unclear and cannot be judged at the moment.

This places you in the rather unpleasant situation of conducting a further comparison against the tools mentioned in

He, D., Choi, A., Pipatsrisawat, K. et al. 2010. Optimal algorithms for haplotype assembly from whole-genome sequence data. Bioinformatics 26, i183–i190.

and

Chen, Z.-Z., Deng, F., and Wang, L. 2013. Exact algorithms for haplotype assembly from whole-genome sequence data. Bioinformatics 29, 1938–1945.

especially the second one in terms of accuracy and recall, and point out the trade-offs between that and your approach.

·

Basic reporting

The basic reporting seems fine now. The authors state clearly that they perform phasing on single individual, short-read datasets, and explain more thoroughly the methodology -- with an added figure -- clearing up any previous confusion. The method seems novel, offering some value to haplotype phasing.

I can only offer some minor comments here, such as cleaning up the grammar/language, for example (sec. 2.4.1), "mergences" is so rarely used that I had to look it up -- it is a synonym of "merging", so I think it would be better to simply say "merging", i.e., "Embedded Merging"

Experimental design

It is now clear that HAHap is focused only on single-individual short-read data. Based on the improved explanation of the methodology, it is clear that HAHap can do much better in this setting of high-coverage noisy short-read data than, e.g., Whatshap, since Whatshap has been designed for low-coverage (also with noise, but from indels) long-read data, with the goal of phasing the largest haplotype blocks as possible. It seems, then, that this comparison -- to Whatshap, maybe HapCut2 as well -- is a bit out of scope.

A possible way to reconcile this, is in addition to computing accuracy (phasing error rate, etc.), is to compute also the _recall_, e.g., quality-adjusted N50 (QAN50) score, see:

Duitama J, et al. Fosmid-based whole genome haplotyping of a HapMap trio child: evaluation of single individual haplotyping techniques. Nucleic Acids Research. 2012; 40:2041–2053.

However, it is certain that short-read data has limited capabilities to obtain good recall (as reflected in good QAN50 scores, for example) -- one of the reasons haplotyping has largely moved to long-reads, since the goal is to assemble long haplotype blocks.

And it is because haplotyping has largely moved to long-reads, that methods optimized for short reads reached its peak some years ago -- this is why methods like Whatshap and HapCut2 are considered state-of-the-art.

So, maybe it is more appropriate to compare against such short-read methods which were very good for their time, for example:

He, D., Choi, A., Pipatsrisawat, K. et al. 2010. Optimal algorithms for haplotype assembly from whole-genome sequence data. Bioinformatics 26, i183–i190.

which is a dynamic programming algorithm (much like Whatshap), but is fixed parameter tractable (FPT) in the read length (Whatshap is FPT in the coverage, hence it dealing with limited coverages) -- allowing it to scale to the 100s-X coverage that Illumina reads have. Another example is:

Chen, Z.-Z., Deng, F., and Wang, L. 2013. Exact algorithms for haplotype assembly from whole-genome sequence data. Bioinformatics 29, 1938–1945.

which is an integer linear programming (ILP) approach that performs _very_ well on short-read datasets.

Validity of the findings

no comment

Additional comments

no comment

Reviewer 4 ·

Basic reporting

The authors addressed the concerns mentioned in the reviews and improved their manuscript.

Parts of the Evaluation Section were rewritten and are clearer now.

Experimental design

The local MEC-based search implemented in HAHap is now described in more detail.
Furthermore, the authors added an example to demonstrate how HAHap differs from MEC-based approaches in noisy regions.

Unfortunately, it is still unclear which objective function the authors attempt to optimize.
They mention that their method solves a minimum spanning tree problem. Likelihoods are used as edge weights. How is a minimum spanning tree helpful if the aim is to maximize the likelihood?
This section is still confusing and requires explanation.

Validity of the findings

Definitions and formulations are clearer now.

The authors used their algorithm for blood group detection based on short reads.
Unfortunately it is not clear what exactly was done in this experiment. The authors should explain in more detail how haplotyping helped to determine subtypes.

Additional comments

Still there is an error when running HAHap on a small Illumina dataset.
"KeyError: 'PG'"

---

## Round 0.3 · accepted · Accept

In my judgment, the manuscript has again substantially improved in clarity, and you have given convincing reasons why it does not make sense to perform certain comparisons. The method is new, and in spite of the comments that phasing based on short reads is slowly becoming obsolete, your method represents an interesting advance of the state of the art and fits well the scope of PeerJ.

#